# Automated Driving with Cooperative Perception Using Millimeter-Wave V2V Communications for Safe Overtaking

**DOI:** 10.3390/s21082659

**Published:** 2021-04-10

**Authors:** Ryuichi Fukatsu, Kei Sakaguchi

**Affiliations:** Department of Electrical & Electronic Engineering, Tokyo Institute of Technology, Tokyo 152-8550, Japan

**Keywords:** 5G, automated driving, cooperative perception, collective perception, V2V communication, millimeter-wave communication, extended sensor

## Abstract

The combination of onboard sensors on vehicles with wireless communication has great advantages over the conventional driving systems in terms of safety and reliability. This technique is often called cooperative perception. Cooperative perception is expected to compensate for blind spots in dynamic maps, which are caused by obstacles. Few blind spots in dynamic maps can improve the safety and reliability of driving thanks to the additional information beyond the sensing of the onboard sensors. In this paper, we analyzed the required sensor data rate to be exchanged for the cooperative perception in order to enable a new level of safe and reliable automated driving in overtaking scenario. The required sensor data rate was calculated by the combination of recognition and vehicle movement to adopt realistic assumptions. In the end of the paper, we compared the required sensor data rate with the outage data rate realized by the conventional V2V communication and millimeter-wave communication. The results showed the indispensability of millimeter-wave communications in automated driving systems.

## 1. Introduction

Vehicles play a key role in modern transportation systems and are indispensable in our daily lives. Market research predicts a continuous growth of the ownership rate of vehicles in the coming years, especially in developing countries. However, as the vehicle penetration rate increases, various social issues arise. Traffic accidents are one of the typical issues. It is said that 93% of traffic accidents are caused by human failure [1]. The lack of public transport for senior citizens in a rural area is also a typical social problem. In Japan, the service frequency of public transport will be halved after the elapsing of 30 years [2]. Since the lack of drivers causes this problem, it is a severe problem in an aging society. Therefore, automated vehicles are expected to make car traffic safer by reducing the impact of human failure and become a new form of transportation in rural areas.

In current Japanese work, automated driving trials are performed to realize Level 2 and 3 automated driving defined by the SAE (Society of Automotive Engineers) [3]. To realize a high level of automated driving, a concept called connected car society is planned. It is designed to connect vehicles to a network so that new services will be started and more advanced automated driving will be realized. The new services can be classified into four groups such as safety, car life support, agent, and infotainment. We focused on safety service among these services to realize Level 4 and 5 automated driving.

One of the biggest challenges of realizing automated vehicles is that automated driving vehicles must recognize the surrounding environment sufficiently in order to detect and localize all objects, obstacles, and pedestrians for safe and effective route planning. Dynamic maps, or HD (High-Definition) maps, are the maps for automated vehicles and are believed to be a key element for safe automated driving. To perform both navigation and collision avoidance, dynamic maps consist of dynamic and static information [4]. Static information provides geographical information that helps an automated vehicle navigate to a destination. On the other hand, dynamic information provides information about surrounding objects that help to prevent a collision. This dynamic information is gained by onboard sensors such as cameras, radar sensors, and LiDAR sensors, which results in a significant hardware cost. Unfortunately, when obstacles block the sensor view, dynamic information has no way to provide information about blind spots. One solution to this problem is connecting an automated vehicle with other sensors, which is called cooperative perception.

The key idea of cooperative perception is to share sensor data obtained from different locations via wireless communication, as shown in Figure 1. It is said that the demands of the data rate for cooperative perception will grow heavily as the level of automated driving grows [5]. This is because the system in an automated vehicle has to recognize the surrounding environment and drive safely by itself. Although many groups have presented the requirements for multiple use cases such as cooperative perception, it is still an open question to find the minimum required sensor data rate to guarantee safe automated driving. Since automated driving roughly consists of obtaining sensor data, processing the sensor data, and reflecting the results of the process in vehicle movement, the requirements for safe automated driving should also consider these processes. Focusing on the sensor data rate, in [6], it was said that an automated vehicle gathers nearly 750 megabytes per second, which will require a high data rate for cooperative perception. Moreover, when raw sensor data are sent for cooperative perception, this high data rate will have a big impact on the performance of wireless communication. From these two facts, it is expected that conventional V2V (Vehicle-to-Vehicle) communications are not enough to support safe automated driving using cooperative perception.

In order to discuss the data rate required for safe automated driving, we focused on overtaking on a two-lane road and derived the required data rate to prevent car-car accidents. This scenario selection was due to the statistics in Japan, which say that head-on collisions account for 50% of accidents in basic road sections [7]. The contributions of this paper consist of the derivation of the requirements that consider safety and showing that millimeter-wave communication has a great ability to share raw sensor data and guarantee safety. In this paper, we adopted a vehicle movement and a recognition process in the required data rate analysis to discuss it under a more realistic assumption. Moreover, we adopted raw sensor data sharing for both processed and raw sensor data sharing for high reliability. In the end, we derived the minimum sensor data rate required to ensure safety. In addition, we proved that millimeter-wave communication is a promising candidate to exchange sensor data among vehicles and allow driving at high speed in the overtaking scenario.

The rest of this paper is organized as follows. Section 2 introduces related works and highlights our contributions. Section 3 explains the system model. Firstly, it introduces cooperative perception and explains the scenario analyzed in this paper. Secondly, it explains the vehicle movement assumed in the scenario and derives the distance to overtake safely. Finally, it explains the process of recognition and derives the required sensor data rate by combining with the vehicle movement. Section 4 shows and compares the safe overtaking realized by the conventional and millimeter-wave V2V communications. In more detail, it explains the V2V channel model, analyzes the antenna space for height diversity, and analyzes the result. Finally, Section 5 concludes this paper.

## 2. Related Works

Our analysis for deriving the minimum data rate required for safe automated driving can be separated into two types of work. The first is about the requirements of the data rate to support V2X (Vehicle-to-Everything) applications. Currently, the requirements of the data rate have been presented by multiple groups. The first group is 3GPP (3rd-Generation Partnership Project). In [8], 3GPP describes requirements for multiple applications such as vehicle platooning, information exchange, and remote driving. In particular, the collective perception of the environment describes the requirements for cooperative perception. This use case considers sharing not only pre-processed data, but also raw data for distributed verification of sensor data. The data rate requirement for pre-processing is 50 Mbps and for raw data is 1 Gbps.

In [5], 3GPP specifies service requirements in six areas, and furthermore, the requirements are described for each level of automation. Cooperative perception can be classified into the area of extended sensors. In the case of a higher degree of automation, the data rate for extended sensors is required to be from 10 to 1000 Mbps.

The second group is 5GAA (5G Automotive Association), who develops end-to-end solutions for future mobility and transportation services. 5GAA defines multiple groups based on the use cases and groups defined by 3GPP and presents requirements in multiple use case scenarios for C-V2X (Cellular-V2X) [9,10,11]. Moreover, 5GAA defines SLRs (Service Level Requirements) that include factors about not only wireless communication performance, but also automotive information (e.g. velocity, positioning, etc.). For example, cooperative perception corresponds to a use case of high-definition sensor sharing that belongs to the group of autonomous driving. In high-definition sensor sharing, the specific data rate is not required, but a max packet size of 1000 bytes is required for processed data, and a larger packet size is required for raw data.

The third group is ETSI (European Telecommunications Standards Institute). In [12], ETSI studies collective perception services that ITS-Ss (Intelligent Transport Systems-Stations) enable to share information about other road users and obstacles that were detected by local sensors. ETSI performs two simulations to get a deeper understanding of collective perception services. In the simulations, radar sensors are assumed as local perception sensors, and processed data such as confidence, geometry, and automotive information are shared. At the end, three factors, which is a trade-off between the generated channel load and the generated awareness, transmission configurations, and message segmentation, are discussed.

As introduced above, 3GPP and 5GAA present the requirements related to cooperative perception. However, both requirements do not have an explicit description of considering safety. On the other hand, ETSI analyzes the performance of collective perception under different traffic scenarios, and a specific message format that includes the processed radar sensor data is defined. Certainly, transmitting the processed data including the reliability of the results rather than transmitting the raw data reduces the channel load. However, the recognition results and the reliability heavily depend on the data transmitter, which will be vulnerable to incidents such as cyber attacks and software errors in the transmitter. Therefore, the option of transmitting raw data by which the receiver can check the results should be also available, which will require a large data rate. In our analysis, the data rate required for cooperative perception was derived by guaranteeing the safety of no collision with other vehicles. Namely, we derived the requirements not by reading the current sensor data rate, but by considering the recognition process and vehicle movement. Moreover, we assumed transmitting raw sensor data, which would require a severe data rate to clarify whether the conventional and the millimeter-wave communications can support it or not.

The second is about clarifying the relation between the safety and performance of wireless communication. In [13], transmitting raw LiDAR sensor data and sharing processed data between vehicles and a mobile edge computing unit were realized by LTE-V2X. Although the authors concluded that there were no negative influences on the automated driving, the driving scenario did not include oncoming vehicles, so that a discussion about safe automated driving was not enough. We also worked with clarifying the relation as follows. In [14], the minimum data rate to overtake was derived, and cooperative perception using millimeter-wave V2V communications contributed to improving safe overtaking. However, the recognition method did not consider the density of lasers from a LiDAR sensor, so that a vehicle would recognize an object even if many lasers were emitted on a very small area of the object. In [15], the minimum data rate required for safe overtaking was derived. However, safe overtaking was discussed only for comfortable braking. In [16], a proof-of-concept of cooperative perception using millimeter-wave communications was performed. It was shown that millimeter-wave communications have the ability of sharing raw sensor data. Although a real-time LiDAR sensor sharing and visualizing blind spots were shown, safety was not analyzed.

In this paper, we adopted edge points as feature points, and the recognition process considered the density of feature points on the visible area. Namely, in order to recognize an object, feature points had to be obtained on the whole area. We also considered the more realistic vehicle movement of overtaking.

## 3. Required Data Rate in V2V for Safe Overtaking

### 3.1. Cooperative Perception and Scenario Description

Once a route destination is decided, automated vehicles calculate a suitable route according to context information, e.g., traffic congestion and toll fees. Information-enriched maps, called dynamic maps, are considered as a promising tool to provide such information to automated vehicles. Dynamic maps consist of static information and dynamic information. In detail, the dynamic information includes information about surrounding moving obstacles such as vehicles and pedestrians, and static information includes HD 3D geographic information, e.g., 3D object distribution and lane information.

Automated vehicles need such dynamic information in order to detect and avoid obstacles. Since dynamic maps must provide various information to perform detection and avoidance, automated vehicles are equipped with various sensors such as radar sensors, stereo cameras, and LiDAR sensors. These sensors provide specific data and information about the vehicle environment and update the dynamic map continuously. LiDAR/radar sensors measure the time it takes for an optical/electrical pulse to return to the LiDAR/radar sensor and provide the distance to the object. Cameras can capture objects’ shapes and movements and even estimate distance by parallax angles with stereoscopic setups. The detailed characteristics of the three sensors are the following:LiDAR sensors provide the distance to an object, and the accuracy is significantly higher than from a radar sensor. Therefore, it can generate a precise 3D map of the surroundings, but it is hard to provide high accuracy data in bad weather, and LiDAR sensors generate a large amount of data.Radar sensors estimate the velocity, distance, and angle of an object and can work in bad weather, but have difficulty providing high-accuracy data.Cameras are good at the classification of objects because they can see color, but the operation is degraded in bad weather or when there is dust in the optical path.

In this paper, we focused on LiDAR sensors in more detail, because these sensors contribute to creating HD dynamic maps, and their output data rate is dominant among automotive sensors, which has a great effect on cooperative perception. According to the measurement principle of LiDAR sensors, when an obstacle is in the FOV (Field-Of-View) of a LiDAR sensor, it can provide the location of the obstacle. On the other hand, it cannot provide any information about an obstacle in a blind spot. Furthermore, as an obstacle that partially blocks a LiDAR sensor’s FOV approaches the LiDAR sensor, the visible area of the LiDAR sensor becomes narrower. As a result, the ability to recognize all obstacles by using only onboard sensors becomes extremely low. To obtain information about blind spots, sharing other sensor data through V2V and V2I (Vehicle-to-Infrastructure) communication is proposed. Sharing sensor data between vehicles is often called cooperative perception. The main advantage of cooperative perception is that it can provide an extended sensing area without substantial additional costs. This additional sensing area contributes to improving traffic efficiency, as well as traffic safety. In [17], it was shown that cooperative perception effectively helped to trigger the early lane changing in the experiment, which contributed to comfortable and safe driving.

The resolution of onboard sensors is an important factor in cooperative perception or recognition. Furthermore, the performance of communication is also an important factor in cooperative perception. Since automated vehicles are equipped with many sensors to recognize their surrounding obstacles, the amount of the generated data is very large, and these data need to be transmitted and processed within the given latency requirements. So far, there are two types of standardized V2V communication systems. One is IEEE (Institute of Electrical and Electronics Engineers) 802.11p, which is one of the dedicated short-range communications. IEEE 802.11p is designed to reduce the latency in V2V communications and consists of two main features. Firstly, it does not require the establishment of a basic service set. Secondly, CSMA/CA (Carrier Sense Multiple Access/Collision Avoidance) is adopted to avoid collision due to simultaneous access. However, the usage of the CSMA/CA mechanism degrades the performance in high traffic density areas mainly due to frequent simultaneous access and many hidden nodes [18]. The other option is LTE (Long-Term Evolution) V2V services. From Release 14, LTE has started to support V2V communications. It has a PC5interface for direct communications and a Uuinterface for long-range cellular network communications. The main advantage of LTE V2V is that it can reuse the same technology for cellular communications. Therefore, we can directly use the already deployed hardware such as base stations. Furthermore, the provided capacity is higher than IEEE 802.11p. However, as the performance of DSRC (Dedicated Short-Range Communication) degrades in high traffic density, the performance of C-V2X also degrades.

Currently, not only conventional V2V communications, but also millimeter-wave communications are expected to be used for cooperative perception. In [19], the authors experimented with cooperative perception without using millimeter-wave communications. Although a 2D LiDAR sensor, which generates fewer data points than a 3D LiDAR sensor, was used in the experiment, position estimation errors were still caused, especially for a high vehicle velocity. This suggests that conventional V2V communications are not sufficient to share HD 3D sensor data. Furthermore, applications for automated driving such as machine learning prefer raw sensor data to compressed sensor data, which leads to requiring a very high data rate [20]. These problems can be solved by using millimeter-wave communications, which provide a high data rate. For example, IEEE 802.11ad has a more than 8 GHz continuous band in four channels, which provides a large channel capacity to transmit sensor data. In [8], as shown in the related works, the requirement for the transmission of raw sensor data rate was estimated as 1 Gb/s. Therefore, we compared the realized safety between conventional and millimeter-wave communications.

Using V2V communications and high-speed information processing technology, automated vehicles are expected to improve traffic efficiency and safety in various driving scenarios. Since considering all driving scenarios makes the analysis complicated, we focused on a driving scenario on a two-lane road where head-on collisions often occur. One of the driving maneuvers on a two-lane road is overtaking. In [21], an overtaking maneuver was included in tactical and operational maneuvers of ADS (Automated Driving Systems), and many Level 4 automated vehicles have the feature of an overtaking maneuver. This fact indicates that, although an overtaking maneuver is riskier than following the leading vehicle, it is necessary to improve traffic flow and shorten the trip time. In human driving, overtaking at a high velocity is very dangerous especially on a road without a lane separator for the oncoming traffic like on a highway.

A human driver is trained to slow down to make space behind the leading vehicle and then try to obtain a clear view of traffic to observe the curvature of the road and closer to the center of the neighbor lane. Once the road ahead is considered safe for overtaking, the driver accelerates and starts the maneuver. With traffic information beyond the limitations of a human FOV from a driver seat, automated vehicles with V2V communication are expected to overtake with less acceleration and deceleration at a high velocity.

To realize safe overtaking, we assumed an overtaking scenario on a two-lane road, as illustrated in Figure 2, and estimated the amount of generated sensor data. This scenario focused on a transition period in which both automated vehicles and human-driven vehicles drive on the road, which limits cooperating vehicles. The driving scenario was that the ego vehicle tries to overtake the blue leading vehicle. Since frequent acceleration and deceleration do not occur on a straight road, the ego and the oncoming vehicles run with the same velocity *V* for simplicity. When the ego vehicle tries to overtake, the blue vehicle drives slow enough for simplicity. Moreover, considering that many Level 4 automated vehicles are equipped with the feature of lane centering, we assumed that all vehicles ran on the center of the road [21]. Using this lane centering function, we also assumed that beam alignment for the V2V communication was ideally performed. When the ego vehicle recognizes the oncoming vehicle by a 3D LiDAR sensor on a roof and safety is not ensured, the ego vehicle does not execute overtaking. The problem is that many beams from the LiDAR sensor are blocked by the leading vehicle. From this point, we call the leading vehicle the blocking vehicle. In this analysis, the blocking vehicle was assumed as an automated vehicle, and the oncoming vehicle was assumed as a human-driven vehicle. Therefore, the ego vehicle can communicate with the blocking vehicle and compensate for the blocked area by cooperative perception. The following sections explain the details of the scenario factors.

### 3.2. Vehicle Movement

In this section, a condition for safe overtaking is discussed from the viewpoint of vehicle movement. In [22], it was said that preventable accidents that can be predicted rationally must not be caused in the ODD (Operational Design Domain) of automated vehicles. From this rule, collision with the oncoming vehicle that can be predicted by automotive sensors should be prevented in the assumed overtaking scenario. To achieve this goal from the viewpoint of vehicle movement, we focused on overtaking movement and braking. The braking movement was considered for an emergency case where the ego vehicle and the oncoming vehicle have to brake during the overtaking. This movement ensures no collision after the braking of both vehicles. The overtaking movement was considered for safe overtaking. This movement ensures no collision with the blocking and oncoming vehicles during the overtaking. The following paragraphs explain the details.

To ensure no collision in an emergency case, we defined the required braking distance. Braking types are classified into the emergency type and comfortable type. When a driver notices an unexpected object on a road, emergency braking occurs with a deceleration of more than 4.5 $m/s^{2}$. Usually, almost all drivers brake with a deceleration of more than 3.4 $m/s^{2}$. This deceleration enables a driver to keep the vehicle in a lane without losing control when braking on a wet roadway. Furthermore, 3.4 $m/s^{2}$ is regarded as being a comfortable rate of deceleration. Comfortable braking is desirable to provide comfortable driving in automated vehicles. The comfortable braking distance is given as follows [23,24].
(1)$$\begin{array}{cc} d_{0} & =0.039\cdot \frac{V^{2}}{3.4} \end{array}$$
where *V* is the velocity (km/h) of a vehicle and $d_{0}$ is the braking distance (m). Considering the reaction time of drivers, the minimum brake reaction times can be 0.64 s for alerted drivers and 1.64 s for an unexpected event [23]. In automated vehicles, electronic control units can perform control in milliseconds. Therefore, the brake reaction time was regarded as negligible. Since both the ego vehicle and the oncoming vehicle have to avoid the collision in this scenario, the required braking distance became $2d_{0}$.

To ensure overtaking movement, firstly, we defined a driving path for overtaking. The driving path is shown in Figure 3 as a black arrow. This driving path was designed to avoid the collision with the blocking vehicle so that the ego vehicle had to turn two times. For example, if we wanted to describe this driving path very simply, it could be described by four quadrants. However, this curve design did not consider vehicle dynamics.
(2)$$\begin{array}{cc} \; & \begin{array}{cc} x(t) & =\int_{0}^{t}\cos\frac{A\theta^{2}}{2}d\theta \\ y(t) & =\int_{0}^{t}\sin\frac{A\theta^{2}}{2}d\theta \end{array} \end{array}$$

The clothoid described as Equation (Equation 2) is one of the curves that considers the vehicle dynamics, where *A* is the clothoid parameter. The clothoid is defined as a trajectory that meets $RL=A^{2}$, where *R* is the radius of curvature and *L* is the length of the curve. Curvature κ of the clothoid can be calculated as κ=At. Curvature and vehicle dynamics relate in terms of vehicle handling. In general, when a vehicle enters a curve, a driver has to turn the steering wheel along the curve. If we make the driving path with four quadrants, a vehicle entering this driving path has to turn the steering wheel quickly. This quick turning is caused by a curvature gap between a straight line and a quadrant. On the other hand, if we use a part of the clothoid from t=0, as shown in Figure 3, the driver does not have to turn the steering wheel quickly, and since κ increases linearly from κ=0, it is enough to turn at a constant velocity.

As explained, the clothoid is suitable to design the driving path in terms of linearly increasing curvature, so it is hard to handle analytically. Therefore, we used the sigmoid curve in our simulation. The characteristics of the sigmoid curve are that it is easier to configure and compute than the clothoid [25].
(3)$$\begin{array}{c} y\left(x\right)=\frac{B}{1+e^{-ax}} \end{array}$$

The function of the sigmoid curve and its parameters B,a are shown in Equation (Equation 3). The way to construct the driving path with the sigmoid curve is shown in Figure 3. In other words, half of the driving path consists of two sigmoid curves that have a mirror symmetry.

In order to configure the sigmoid curve, we needed to determine the $B,\;a,\;x_{0}$ parameters. The parameter *B* depends on the road width. In this driving path, the ego vehicle was assumed to move from the center of the lane to the center of the neighbor lane, then return to the first lane. From this assumption, *B* is equal to the width of a single lane. The parameter *a* determines the curvature of the sigmoid curve. In order to determine *a*, we considered a slip and constructed a sigmoid curve that does not cause slip at a minimum curvature radius shown in Figure 4. The judgment of the slip is performed by the following formula.
(4)$$\begin{array}{cc} \frac{mv^{2}}{R}⋚\mu mg & \left\{\begin{array}{c} \mathrm{slip}:\;\frac{mv^{2}}{R}>\mu mg\\ \mathrm{safe}:\;\mathrm{otherwise} \end{array}\right. \end{array}$$
where *m* is the mass of the vehicle, *v* is the velocity of the vehicle, *R* is the curvature radius at the point where the vehicle places, μ is the coefficient of static friction, and *g* is gravity acceleration. Since we assumed that the ego vehicle drives along the path at a constant velocity, the minimum curvature radius without slip is $R_{\min}=v^{2}/\mu g$ from Equation (Equation 4). Finally, the parameter $x_{0}$ determines the length of the sigmoid curve. The length is determined by the duration to complete the overtaking, which was set in advance.

Figure 5 shows the examples of the driving path at 20 km/h and 50 km/h. From the figure, it is shown that when the vehicle velocity becomes low, the slope of the driving path becomes steep. This can be explained by the definition of $R_{\min}$. In other words, a vehicle driving at a low velocity can turn sharply without slipping.

To compare with the distance required by the braking, firstly, we derived the distance for the overtaking driving path. Since the oncoming vehicle drives on the neighboring lane, the ego vehicle has to finish overtaking by the time the oncoming vehicle arrives at the collision point.

Figure 6 shows both driving paths from the start point to the collision point, where *v* is the velocity of the vehicle, $t_{o}$ is the duration to complete the overtaking, and $x_{0}$ is shown in Figure 4. The collision occurs when the ego vehicle moves to the center of the neighboring lane to overtake the blocking vehicle. Since the driving path is a mirror symmetry curve, when the ego vehicle arrives at the center of the neighbor lane, the oncoming vehicle moves for $t_{o}/2$. Therefore, the distance required for the overtaking driving path becomes $vt_{o}/2+2x_{0}$.

From the above discussion, we combined the distance required for the driving path and the comfortable braking. Figure 7 shows the distance required for comfortable braking and for the three driving path cases. It is shown that the driving path required a larger distance than the comfortable braking distance at a low velocity, and this was reversed at a high velocity. Therefore, considering the driving path is important especially at a low velocity. Namely, the required distance $d_{\mathrm{req}}$ can be formulated as follows.
(5)$$\begin{array}{c} d_{\mathrm{req}}=\max\left(2\times 0.039\times \frac{V^{2}}{3.4},\;\frac{vt_{o}}{2}+2x_{0}\right) \end{array}$$

### 3.3. Derivation of the Required Data Rate

In this section, the details of the recognition process are introduced. In general, object recognition can be classified into two cases. One is specific object recognition. This recognition tries to classify an object as a specific object. The other is general object recognition. In contrast to the former recognition, this recognition tries to classify an object as a generic object. Since we focused on the recognition of a vehicle, specific object recognition was adopted, and we refer to the recognition target as the target vehicle. In this case, the ego vehicle wants to prevent a collision with the oncoming vehicle so that the oncoming vehicle becomes the target vehicle. The recognition part consists of three phases. The first phase is the simulation of LiDAR sensor data in the virtual environment and clustering point cloud about the target vehicle. The second phase is the extraction of feature points from the clustered points. The final phase is the decision about recognition. The following paragraphs explain the details of each phase.

In the first phase, regarding lasers from a LiDAR sensor as geometric optics, ray-tracing simulation of LiDAR sensor data was adopted. In order to implement ray tracing easily, objects such as vehicles, buildings, and roads consist of triangle meshes. From this setting, a point p on a triangle mesh can be described with three position vectors $p_{1}$, $p_{2}$, and $p_{3}$ and two parameters u,v as the following formula.
(6)$$\begin{array}{c} p=\left(1-u-v\right)p_{1}+up_{2}+vp_{3} \end{array}$$

Furthermore, the point p can be also described by a normalized direction vector d departing from the laser source O to p.
(7)$$\begin{array}{c} p=O+td \end{array}$$

Since the laser propagates in three-dimensional space, the departure angle can be described by azimuth angle ϕ and elevation angle θ. When a point is on the mesh, parameters *u* and *v* have to meet 0≤u,v≤1 and 0≤u+v≤1. On the other hand, parameter *t* has to meet 0≤t. In order to confirm whether these conditions are met or not, we solved these parameters by combining Equations (Equation 6) and (Equation 7) and adopting Cramer’s rule.

As mentioned above, before extracting the feature points only from the target vehicle points for recognition, clustering is needed to remove irrelevant points. In this simple ray tracing algorithm, the function of linking the hit object to the laser is implemented. As a result, the LiDAR sensor in our simulation knows which object the laser is reflected from so that we can only select the points of the target vehicle and perform clustering easily.

In the second phase, we extracted feature points from the clustered points. When we want to describe features of point cloud data, or LiDAR sensor data, a feature descriptor is often used. SHOT (Signature of Histogram of OrienTation) and PFH (Point Feature Histogram) are the typical feature descriptors. These descriptors use a histogram to describe features around a point [26,27]. In general, the calculation time of a feature descriptor depends on the dimension of the descriptor. In order to avoid this complicated discussion, we used edge points, which are basic features. Extracting edges was performed by PCA (Principal Component Analysis) [28]. This PCA method is faster and more robust to noise than using a Gauss map. The key point of this process is that edge points are extracted by the eigenvalues of a covariance matrix. The quantity made of the eigenvalues is called the surface curvature, and it is calculated for each point. When the surface curvature exceeds a threshold, the point is regarded as an edge point. The threshold is tuned by observing the distribution of the surface curvature.

The final phase is the decision about recognition. In this simulation, we adopted model-based recognition. This recognition method is a matching problem between scene and model points. Scene points are obtained from the output of the LiDAR sensor. On the other hand, model points are prepared in advance and have enough points to extract the feature points of the target vehicle. The process of this recognition consists of calculating the feature points of the model and scene points and searching for the correspondence of the feature points between the model and scene points. If there are corresponding points, clustering with regard to the corresponding points is performed.

We simplified two points about this model-based recognition process. The first point is using not the entire scene points, but the points clustered from the scene points. This extraction is performed in the first phase of ray tracing. The second point is the decision way of recognition. We defined a recognition score *S* as the ratio of the number of edge points shown in Equation (Equation 8). $N_{\mathrm{get}}$ and $N_{\mathrm{LOS}}$ are the number of edge points calculated from the two configurations, as shown in Figure 8.
(8)$$\begin{array}{c} S=\frac{N_{\mathrm{get}}}{N_{\mathrm{LOS}}} \end{array}$$

The difference of these configurations is that the right configuration includes all objects, but the left configuration only includes the target vehicle, as shown in Figure 8. In the Figure 8a case, the sensing range of the LiDAR sensor for the ego vehicle is described by the green range. This environment enables the ego vehicle to sense the target vehicle with an LoS (Line-of-Sight). In the Figure 8b case, there are two LiDAR sensors. One is on the ego vehicle, but contrary to the former case, the blue vehicle blocks the sensing, as shown by the yellow range. The other is on the blue blocking vehicle, which senses with an LoS the same as the former case. The edge points obtained in the Figure 8a case are regarded as the maximum number of edge points of the target vehicle that the ego vehicle can obtain. On the other hand, the edge points in Figure 8b can be obtained in two ways, that is using cooperative perception or not. Using cooperative perception, the edge points calculation is based on the yellow and blue sensing range, while without cooperative perception, it is only based on the yellow sensing range. Figure 9a shows the entire edge points in the model points. Figure 9b shows two points. One is the white edge points obtained under the Figure 8a configuration, and the other is the red points obtained under the Figure 8b configuration using cooperative perception. Since the red points are also obtained from an LoS place, the red and white points’ distribution is similar.

Counting the number of edge points is different between the two configurations. $N_{\mathrm{LOS}}$ in Equation (Equation 8) is the number of LoS edge points obtained in the Figure 8a case. In detail, firstly, the LoS edge points of the model points are calculated by PCA edge extraction and voxelized. The resolution of the voxelization is based on the error range of a LiDAR sensor. Secondly, the edge points are moved and aligned with the target vehicle. Finally, the voxelized edge points that are in the LoS from the ego vehicle are extracted. On the other hand, the first process for $N_{\mathrm{get}}$ is the simulation of the LiDAR sensor data under the Figure 8b configuration. Since we focused on whether the scene points are on edges or not, we defined that the scene points have information about one feature point when a scene point is near the voxelized edge points of the model points. As a result, $N_{\mathrm{get}}$ is the total number of voxelized edge points obtained from the scene points. After the calculation of $N_{\mathrm{LOS}}$ and $N_{\mathrm{get}}$, the ratio and threshold were compared, and when the ratio was more than the threshold, we defined that the ego vehicle recognized the target vehicle.

In the vehicle movement part, we derived the required distance $d_{\mathrm{req}}$ to avoid a collision. Furthermore, from the recognition process, we can judge that the ego vehicle recognizes the target vehicle at a given distance. Therefore, the combination of the required distance $d_{\mathrm{req}}$ in Equation (Equation 5) and the recognition process derives the required sensor data rate $R_{\mathrm{req}}$ to avoid a collision as follows.
(9)$$\begin{array}{cc} \left\{\bar{r_{\phi }},\;\bar{r_{\theta }}\right\} & =\underset{\left\{r_{\phi },\;r_{\theta }\right\}}{\arg\;\min\;}S\left(r_{\phi },\;r_{\theta }|d_{\mathrm{req}},\;d_{\mathrm{be}}\right)>0.9 \end{array}$$
(10)$$\begin{array}{cc} R_{\mathrm{req}} & =\left(\left\lfloor \frac{A_{\phi }}{\bar{r_{\phi }}}\right\rfloor +1\right)\times \left(\left\lfloor \frac{A_{\theta }}{\bar{r_{\theta }}}\right\rfloor +1\right)\times F_{\mathrm{scan}}\times D_{\mathrm{symbol}} \end{array}$$
where *S* is the recognition score in Equation (Equation 8), $A_{\phi }$ and $A_{\theta }$ are the scanning range in the azimuth and elevation angle, $F_{\mathrm{scan}}$ is scan frequency (Hz) of the LiDAR sensor, and $D_{\mathrm{symbol}}$ is the amount of information per one laser point (bit). Note that the required sensor data rate depends on the velocity *v* of the ego vehicle and the distance $d_{\mathrm{be}}$. Finally, we can derive the realized maximum overtaking velocity by obtaining the minimum outage capacity that exceeds the required sensor data rate, which will be introduced in the simulation section.

Figure 10 shows the required sensor data rate with the two options such as cooperative perception and driving path. The solid (dotted) line with square markers shows the minimum required sensor data rate to overtake with (not) using cooperative perception and not considering the driving path. The solid (dotted) line with circle markers considers the driving path with (without) cooperative perception. From the figure, firstly, we can see that all required sensor data rates rapidly increased. This rapid increase was due to the laser density, or the resolution of the LiDAR sensor, which became rapidly sparse at a far place. In the case of no cooperative perception, since the blocking vehicle interrupted the sensing, a much higher resolution was required so that the required sensor data rate increased rapidly. Secondly, there was a difference between considering the driving path or not. This reflects the result of the 5 s driving overtaking shown in Figure 7 so that no difference was seen at more than 60 km/h.

Figure 11 shows the required sensor data rate with $d_{\mathrm{be}}=5,\;10,\;15$ m and 5 s overtaking. In the case of using cooperative perception, as $d_{\mathrm{be}}$ becomes larger, the required sensor data rate becomes smaller. When $d_{\mathrm{be}}$ is large, the blocking vehicle gets near to the oncoming vehicle. This allows the blocking vehicle to recognize the oncoming vehicle with a low-resolution LiDAR sensor. On the other hand, the required sensor data rate in no cooperative perception depends on two factors, which leads to a complicated result. One is the distance $d_{\mathrm{oe}}$. When $d_{\mathrm{oe}}$ is large with the presence of the blocking vehicle, it is easy for a high-resolution LiDAR sensor on the ego vehicle to see the shape of the whole oncoming vehicle in a small sensing range, which obviously has a limit for the recognition. The other is distance $d_{\mathrm{be}}$. As the blocking vehicle gets near to the ego vehicle, the blocking vehicle blocks a large part of the range that sees the oncoming vehicle except for a very near location. Since the LiDAR sensor is on the roof, a large part of the blocking vehicle does not block the sensing in the case of a very near location. From Figure 11, the required sensor data rate becomes high from $d_{\mathrm{be}}=15$ m to $d_{\mathrm{be}}=10$ m, but it becomes low from $d_{\mathrm{be}}=10$ m to $d_{\mathrm{be}}=5$ m. This result tells us that sensing with no cooperative perception on a two-lane road heavily depends on many factors such as the size and the location of vehicles, which will make the requirements complicated. On the other hand, sensing with cooperative perception simply depends on $d_{\mathrm{be}}$.

Figure 12 shows the required sensor data rate using two different LiDAR sensors. One is a linear spacing LiDAR sensor, and the other is a non-linear spacing LiDAR sensor. In the case of linear spacing, the LiDAR sensor has an equally spaced elevation angle resolution such as Velodyne VLP-16. On the other hand, a non-linear spacing LiDAR sensor such as Velodyne VLP-32 has a dense and sparse spacing part. In this analysis, we fixed the number of lasers between the two LiDAR sensors. The details of non-linear spacing are shown in Table 1. From the figure, we can see that a non-linear spacing LiDAR sensor has a better ability to recognize a far object. However, notice that non-linear spacing provides sparse information about a near object.

## 4. Performance Evaluation of Millimeter-Wave V2V

### 4.1. Millimeter-Wave V2V Communications with Antenna Height Diversity

In the current V2X (Vehicle-to-Everything) communication system, DSRC (Dedicated Short-Range Communication) is the most popular V2X communication system. It is natively designed to support communication with high-speed vehicles. However, since it uses 5.8 or 5.9 GHz, which provides a several Mbps transmission rate, it has difficulty supporting cooperative perception. Moreover, when the density of vehicles becomes high, the performance becomes rapidly degraded due to CSMA/CA. On the other hand, C-V2X (Cellular-V2X) was developed by 3GPP (3rd-Generation Partnership Project) to enhance ITSs (Intelligent Transport Systems) and support automated driving. The main characteristics of C-V2X are that the Uu and PC5 interfaces are prepared for different use cases. When a user wants to access infrastructure such as an application server, the Uu interface is used. The PC5 interface is prepared for direct communications between the users. This direct communication can be utilized for cooperative perception. Although the C-V2X function of scheduling can avoid communication collision, the data rate of PC5 is not enough for cooperative perception due to it using 5.9 GHz [29].

As introduced above, the conventional V2X communications are not suitable for cooperative perception in terms of the data rate. Therefore, new V2X communication systems such as IEEE 802.11bd and NR-V2X (New Radio-V2X) are currently under development. Both IEEE 802.11bd and NR-V2X plan to include millimeter-wave communications to support sending high-resolution 3D maps [30]. This use case can be realized by millimeter-wave communications, which provide high throughput. However, millimeter-wave has a greater path loss and is more easily blocked than the frequency used in conventional V2X communications. To compensate for these defects, strong directivity and ensuring LoS are necessary. From a different point of view, these defects can be advantages for spatial re-usability. In other words, blockage effects and strong directivity reduce interference to surrounding vehicles [31]. In this paper, we compared the outage capacity of millimeter-wave communication and conventional communication and estimated how much this contributed to safe driving.

In this paragraph, the V2V (Vehicle-to-Vehicle) channel model for moving vehicles is introduced. In [32,33], the authors measured 5 and 60 GHz and performed modeling of the measured data. As a result, it was shown that the two-ray ground reflection model was suited for the V2V channel. Therefore, we adopted the two-ray ground reflection model as a large-scale path loss model in the simulation, as shown in Figure 13. The additional characteristics of this model are that vibrating of both the transmitter and receiver due to vehicle movement has an effect on small-scale fading [33]. To avoid the fading in the driving environment, in [34], it was shown that, when one vehicle with multiple receivers is chasing the other vehicle with a transmitter, diversity gain is maximized by a vertically displaced antenna rather than a horizontally displaced antenna. In [35], the authors analyzed the outage capacity under the 60 GHz two-ray ground reflection model that follows the Rayleigh and the Rice distribution. They derived theoretically that height diversity provides large improvements rather than horizontal space diversity. In [36], the author derived that the antenna space for height diversity should be more than 10 cm. In our analysis, we assumed height diversity at the receiver and discussed how much height diversity improved the outage capacity. Since 99.99% reliability is required in cooperative perception, we estimated the improvement by a 0.01% outage capacity [8]. Moreover, we derived the best antenna space that improved the outage capacity among 5, 30, and 60 GHz.

Firstly, we analyzed the basic characteristics of this channel model. As described above, this channel model can be separated into the effect of two-ray ground reflection and antenna vibration. The received power $P_{r}$ under the two-ray ground reflection is formulated as follows.
(11)$$\begin{array}{c} P_{r}=\frac{P_{t}}{L(r_{d})}{\left|\sqrt{G_{d}}\left(\frac{\lambda }{4\pi r_{d}}\right)+\sqrt{G_{r}}\left(\frac{\lambda }{4\pi r_{r}}\right)\Gamma e^{-j\{k\left(r_{d}-r_{r}\right)\}}\right|}^{2} \end{array}$$
where $P_{t}$ is the transmission power, $G_{d}$ and $G_{r}$ are the antenna gains for direct and reflected waves, $r_{d}$ and $r_{r}$ are the optical path length for direct and reflected waves, $L(r_{d})$ is the absorption factor at 60 GHz by oxygen as 15 dB/km, λ is the wavelength, *k* is 2π/λ, and Γ is the complex reflection coefficient. When the antenna vibration caused by the motor on the vehicle is adopted in this channel model, it changes $r_{d}$ and $r_{r}$. In [33], the authors modeled this antenna vibration by a Gaussian distribution $N_{r}\left(0,\;\sigma_{0}^{2}\right)$ where $\sigma_{0}$ is 0.0319 m. This vibration causes a shift of all fading points, and all receiving places have the possibility to encounter strong fading. To avoid strong fading, height diversity was adopted. In this case, the receivers vibrated by the same motor so that they followed the same distribution as Equations (Equation 12) and (Equation 13).
(12)$$h_{\mathrm{tv}}=h_{t}+\delta_{t}\;\mathrm{where}\;\delta_{t}∼N_{t}\left(0,\;\sigma_{0}^{2}\right)$$
(13)$$\left.\begin{array}{c} h_{\mathrm{rv}1}=h_{r1}+\delta_{r}\\ h_{\mathrm{rv}2}=h_{r2}+\delta_{r} \end{array}\right.\;\mathrm{where}\;\delta_{r}∼N_{r}\left(0,\;\sigma_{0}^{2}\right)$$

Finally, the 0.01% outage capacity $C_{\mathrm{out}}$ was calculated under the height diversity.
(14)$$\begin{array}{c} P(c<C_{\mathrm{out}})=0.01\% \end{array}$$

Figure 14 shows the above discussion in the case of 60 GHz. The black solid line shows the basic characteristic of the two-ray ground reflection model. The blue dotted and dashed lines show the moment of vibrating to 3σ and −3σ. We can see that the strong fading points are shifted to the left and right. The red dotted line shows the 0.01% outage capacity under no height diversity. We can see that the fading occurred at an arbitrary V2V distance. In particular, there were sharp drops at 18 and 35 m, and there was a sharp rise at 84 m. These sharp changes were due to antenna vibration, which made the fading point at 57 m move ±21 m and at 28 m move ±10 m. Considering that a shift from −3σ to +3σ happened at around 99.7% and the shift of 0.01% outage capacity was larger than the shift of ±3σ, this shift was a reasonable result. The red solid line shows the outage capacity with height diversity. Although it was improved from no diversity, the outage capacity gradually changed up and down. The main reason for this change was that there were some places where both receivers encountered strong fading.

To improve this outage capacity with height diversity, we had to solve the changing up and down of the outage capacity. The following analysis consisted of two parts. The first analysis focused on the best receiving antenna space for the second receiving antenna. In this analysis, carrier frequency, inter-vehicle distance, the height of the transmitting antenna, and the height of the first receiving antenna were given. The second analysis focused on the relation between the communication range and the number of receiving antennas. In the analysis of the receiving antenna space, firstly, we focused on the difference of the phase difference between the direct wave path and the reflected wave path. When the height diversity worked well, this phase difference was around π, as the following formula.
(15)$$\begin{array}{cc} \; & \frac{2\pi }{\lambda }\left\{\left(r_{d2}-r_{r2}\right)-\left(r_{d1}-r_{r1}\right)\right\}\equiv \pi \;\left(\mathrm{mod}\;2\pi \right) \end{array}$$
where $r_{d1}$ ($r_{d2}$), $r_{r1}$ ($r_{r2}$) are the lengths of the direct and reflected paths from the transmitter whose height is $h_{\mathrm{tv}}$ to the lower (upper) receiver whose height is $h_{\mathrm{rv}1}$ ($h_{\mathrm{rv}2}$). Using the approximation of $\sqrt{1+x}\approx 1+x/2\;\left(1\gg |x|\right)$, Equation (Equation 15) can be described as follows.
(16)$$\begin{array}{cc} \; & \frac{2\pi }{\lambda }\left\{\left(r_{d2}-r_{r2}\right)-\left(r_{d1}-r_{r1}\right)\right\}\\ \; & \approx -\frac{4\pi }{d\lambda }\;h_{\mathrm{tv}}\left(h_{\mathrm{rv}2}-h_{\mathrm{rv}1}\right)\\ \; & =-\frac{4\pi }{d\lambda }\left(h_{t}+\delta_{t}\right)\left(h_{r2}-h_{r1}\right)\;\left(=\Phi (\delta_{t})\right) \end{array}$$
where *d* is the inter-vehicle distance. From the above approximation, it was shown that variable in the difference was only $\delta_{t}$. We expressed the difference as $\Phi (\delta_{t})$. When there is no vibration, that is δ=0, we describe the solutions $h_{n}$ of Equation (Equation 16) as follows.
(17)$$\begin{array}{c} h_{n}=h_{r2n}-h_{r1n}\;\mathrm{such}\;\mathrm{that}\;\Phi \left(0|h_{n}\right)=-\frac{4\pi }{d\lambda }\left(h_{t}+0\right)h_{n}=\left(2n+1\right)\pi \end{array}$$

To choose the best $h_{n}^{\mathrm{best}}$ among the solutions $h_{n}$, we estimated Φ where $\delta_{t}$ ranged from $-3\sigma_{0}$ to 3σ, which fell within around 99.73%. This was because, recalling that the goal was to solve the changing up and down of the outage capacity due to the antenna vibration, it was necessary to select one solution for which the phase difference Φ did not change more than π/2 with a large probability. Since $\Phi \left(0|h_{n}\right)-\Phi \left(-3\sigma_{0}|h_{n}\right)$ and $\Phi \left(3\sigma_{0}|h_{n}\right)-\Phi \left(0|h_{n}\right)$ were the same, we focused on $\Phi \left(0|h_{n}\right)-\Phi \left(-3\sigma_{0}|h_{n}\right)$. Moreover, we restricted the maximum phase difference to 65°, which ensured more than half power at the other receiver.

The second analysis was about the communication range. In the first analysis, we analyzed the best receiving antenna space under the fixed inter-vehicle distance. However, it was not realistic to equip receiving antennas for each inter-vehicle distance, which would require too many antennas. In order to solve this problem, we derived the range of the inter-vehicle distance where the height diversity worked well and minimized the number of receiving antennas. Under the $h_{n}^{\mathrm{best}}$, we defined the valid communication range *R* as the range for which the difference of Φ was from 115° to 245°, as follows.
$$\begin{array}{cc} R_{\min} & =\frac{4\pi }{2\pi \frac{180+65}{360}\lambda }h_{t}h_{n}^{\mathrm{best}} \end{array}$$
(18)$$\begin{array}{cc} \; & =\frac{144}{49\lambda }\;h_{t}h_{n}^{\mathrm{best}}\\ R_{\max} & =\frac{4\pi }{2\pi \frac{180-65}{360}\lambda }h_{t}h_{n}^{\mathrm{best}} \end{array}$$
(19)$$\begin{array}{cc} \; & =\frac{144}{23\lambda }\;h_{t}h_{n}^{\mathrm{best}} \end{array}$$

From Equations (18) and (19), as the antenna space $h_{n}^{\mathrm{best}}$ became large, the minimum and maximum effective communication distance became large. Furthermore, as the wavelength λ became short, these distances became also large.

For example, Figure 15 plots the several cases of the 0.01% outage capacity at 60 GHz. The black line shows the outage capacity without height diversity. The red and blue lines show the outage capacity with a 10 and a 20 cm antenna space. The red and blue two-headed arrows show the effective communication range in the case of a 10 and a 20 cm antenna space. From Figure 15, we can see that the outage capacity in the effective range was better than the outside. Although there are several solutions to Equation (Equation 15) at a 10 cm antenna space in d<22m and at a 20 cm antenna space in d<48m, rapid fluctuation of the difference of the phase difference degraded the outage capacity. On the other hand, since the difference became stable at more than π in d>48m at a 10 cm antenna space and in d>95m at a 20 cm antenna space, the improvement from no height diversity gradually decreased.

From the above analysis, we concluded that the antenna space determined short-range or long-range communication. The statistics show that the average speed on highways is about 70 km/h and on roads is about 30 km/h [37]. Moreover, we adopted the two second rule to decide the average V2V distance. From this rule, we derived the average distance on roads, which was 17 m, and on highways, which was 39 m. Based on these average distances, we proposed two additional receivers for height diversity that supported the above two values. At 60 GHz, we set the first receiver at a 5 cm space whose effective range was from 11 m to 24 m and the second receiver at a 12 cm space whose effective range was from 27 m to 57 m. In the case of 30 GHz, ten centimeters and 24 cm of space were necessary. Finally, in the case of 5 GHz, sixty centimeters and 144 cm of space were necessary. Figure 16 shows the outage capacity at 5, 30, and 60 GHz with height diversity. Note that 5, 30, and 60 GHz used three receivers as proposed in the above discussion. We can see that all outage capacities had no sharp drop and decreased linearly.

### 4.2. Performance of Millimeter-Wave V2V Communications to Support Safe Overtaking

To estimate the amount of the minimum required sensor data for safe overtaking, we performed a simulation. The required sensor data rate was derived with $d_{\mathrm{be}}$ and $d_{\mathrm{oe}}$. Figure 17a shows the process flow of the simulation. Firstly, the output of the LiDAR sensors on the blocking and ego vehicle was simulated. When the ego vehicle used cooperative perception, it could use not only its sensor data, but also the sensor data of the blocking vehicle for the recognition process. In the recognition process, we defined that if the recognition score described in Equation (Equation 8) was more than 0.9, the ego vehicle recognized the oncoming vehicle. The recognition score was calculated under the prepared LiDAR sensor resolution sets of $\left(r_{\phi },\;r_{\theta }\right)$. If the ego vehicle failed to recognize the oncoming vehicle, it believed that there were no vehicles on the oncoming lane. Since the ego vehicle did not know exactly whether there were vehicles on the oncoming lane, this misunderstanding led to a collision. If the ego vehicle recognized the oncoming vehicle, then it additionally checked two factors. One was ensuring a comfortable braking distance for preventing a collision. The other was ensuring the driving path for overtaking. In this simulation, the required time for overtaking was set to 5 s. If the ego vehicle did not pass either, it would stay on the lane. If it passed, it could overtake the blocking vehicle.

Figure 18, Figure 19 and Figure 20 show the simulation result of $d_{\mathrm{be}}=5,\;10,\;15$ m under the parameters of Table 1. The horizontal axis denotes the velocity of the ego vehicle, and the vertical axis denotes the sensor data rate required for the safe overtaking and the outage capacity of each carrier frequency. The black solid and dotted lines with markers show the required sensor data rate with and without cooperative perception. Since the calculation time grew drastically at more than 8 Gbps, extrapolation was used. The green, red, and blue solid lines show the realized 0.01% outage data rate for each $d_{\mathrm{be}}$ at 5, 30, and 60 GHz. Figure 21 shows the result of $d_{\mathrm{be}}=10$ m using non-linear spacing LiDAR sensors.

Considering the realized data rate and the required sensor data rate using cooperative perception, the maximum velocity for safe overtaking under $d_{\mathrm{be}}=5,\;10,\;15$ m at 60 GHz was 66, 64, 67 km/h and at 30 GHz was 51, 49, 54 km/h. Since the realized data rate at 5 GHz was too small, the maximum velocity for safe overtaking was less than 20 km/h in all cases. In all $d_{\mathrm{be}}$ cases, cooperative perception using 60 GHz constantly ensured around 65 km/h for safe overtaking. In the case of using non-linear spacing LiDAR sensors, when 5, 30, and 60 GHz were used for cooperative perception, the ego vehicle could safely overtake 0, 9, and 7 km/h faster than using linear spacing LiDAR sensors. Although non-linear spacing LiDAR sensors improved the overtaking, the effect of providing sparse information about near objects should be noticed, especially in other driving scenarios.

For the final discussion, we compared these results with the current requirements for cooperative perception. In [5], one-thousand megabits per second and 10 ms of max end-to-end latency were required for a higher degree of automation to prevent an imminent collision by extended sensors or cooperative perception, which allowed the maximum sensor data rate at 100 Mbps. On the other hand, in [8], one gigabit and 3 ms of end-to-end latency were required for collective perception of environment or cooperative perception, which allowed the maximum sensor data rate at 3 Mbps. In our analysis, even if an automated vehicle drives at around 30 km/h, around a 1 Gbps sensor data rate is required for safe driving. From the above comparison, although we considered only LiDAR sensors for recognition, considering safety affected the requirements.

These results show that millimeter-wave communication has a big potential to contribute to safe overtaking and smooth traffic. Although the actual data rate will be lower than these outage data rates, millimeter-wave communication especially at 60 GHz has a large margin. Therefore, we concluded that millimeter-wave communication has the ability to perform safe overtaking at a high velocity, and considering the actual data rate, sixty gigahertz would be a promising frequency.

## 5. Conclusions

The contribution of this paper focused on two aspects. Firstly, we derived the sensor data rate required for the safe overtaking by considering the driving path and comfortable braking and showed that as the velocity became higher, the required sensor generated data rate increased drastically. At a low velocity, the effect of considering the driving path became dominant, and at a high velocity, comfortable braking became dominant. Secondly, techniques for cooperative perception with 30 and 60 GHz millimeter-wave communication made it possible to safely overtake at a high velocity such as around 50 and 65 km/h due to the availability of sharing a large amount of sensor information in real time. From this analysis, we concluded that, considering the actual data rate, using 60 GHz for cooperative perception is a promising way to perform safe overtaking at a high velocity. Finally, since the recognition process is classical, future tasks are the adoption of global or local descriptors for the recognition process.

## Figures and Tables

**Figure 1 sensors-21-02659-f001:**
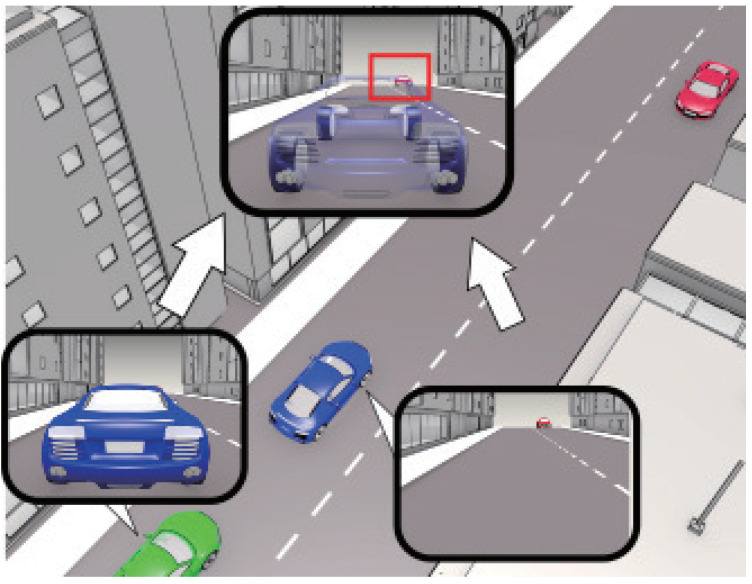
An illustration of the concept of cooperative perception. Although the red oncoming vehicle is invisible to the green ego vehicle due to the blue vehicle blocking it, the cooperative perception between the ego vehicle and blocking vehicle changes it to visible.

**Figure 2 sensors-21-02659-f002:**
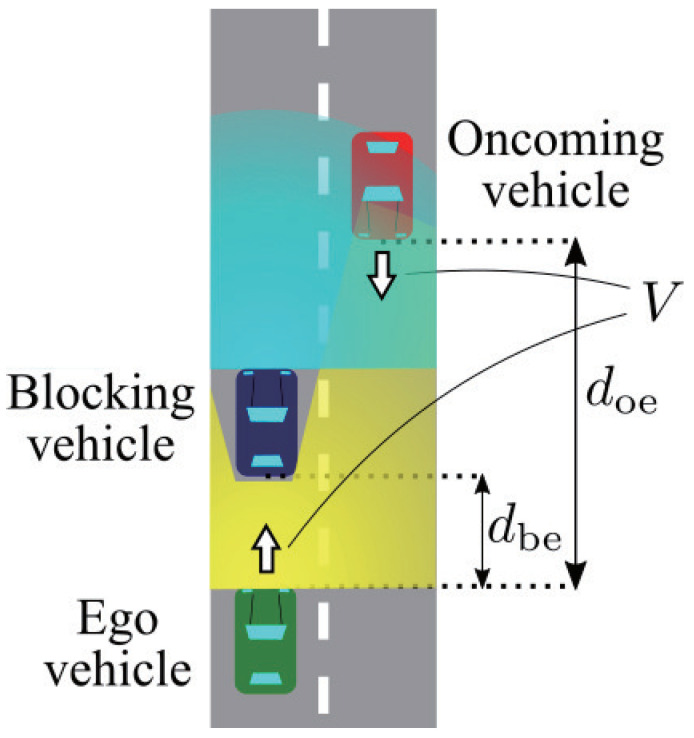
Illustration of the ego vehicle equipped with a 3D LiDAR sensor trying to execute overtaking. The yellow region is the sensing region of the ego vehicle’s LiDAR sensor.

**Figure 3 sensors-21-02659-f003:**
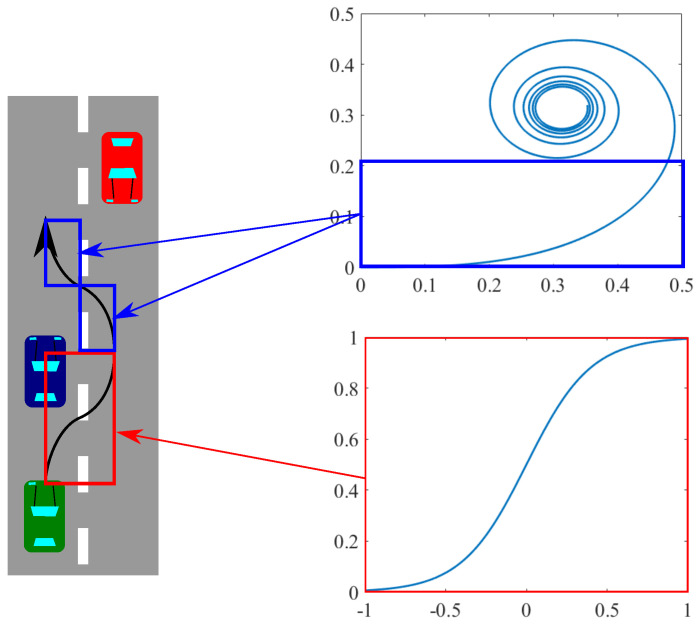
The driving path of overtaking and the two types of curves for approximation.

**Figure 4 sensors-21-02659-f004:**
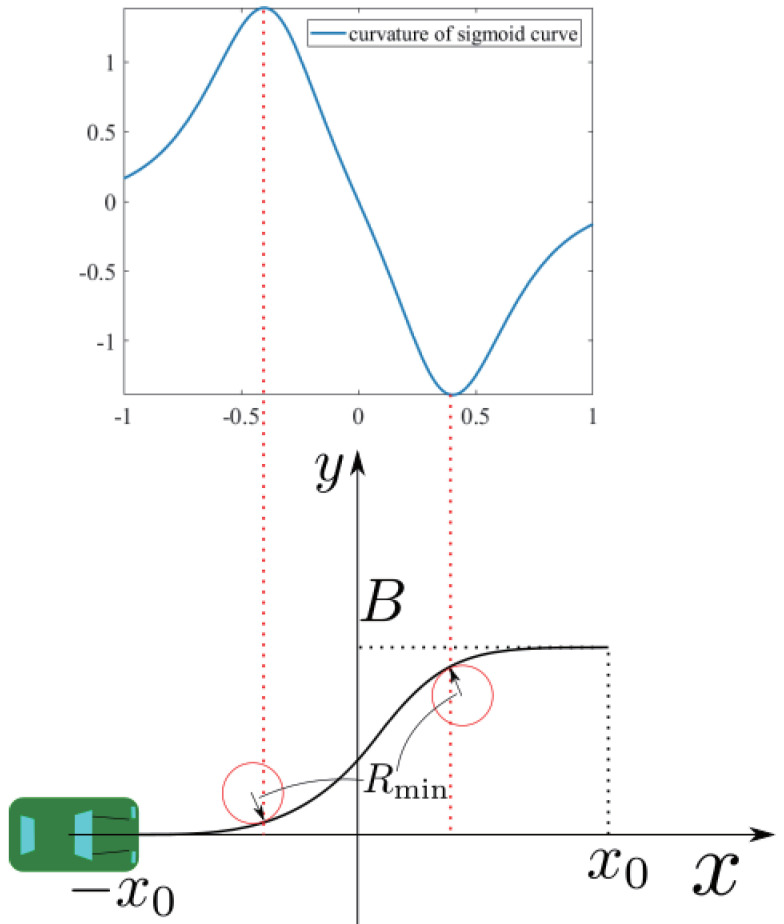
The parameters for the sigmoid curve under a=5 and the location of the minimum curvature radius.

**Figure 5 sensors-21-02659-f005:**
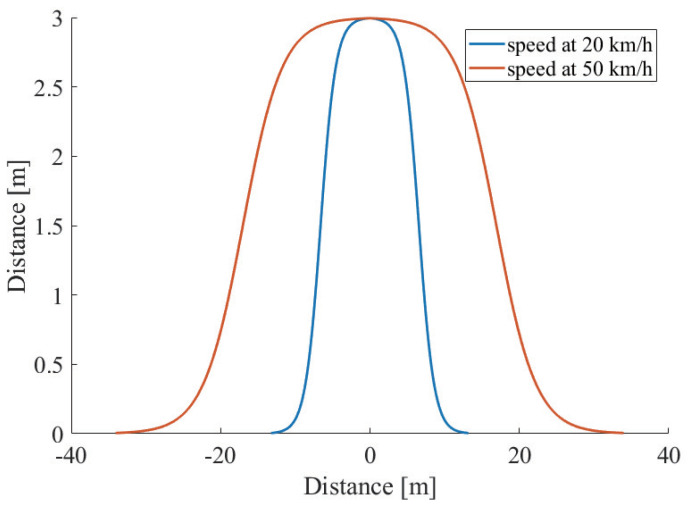
The examples of two driving paths where the duration to complete the overtaking is 5 s.

**Figure 6 sensors-21-02659-f006:**
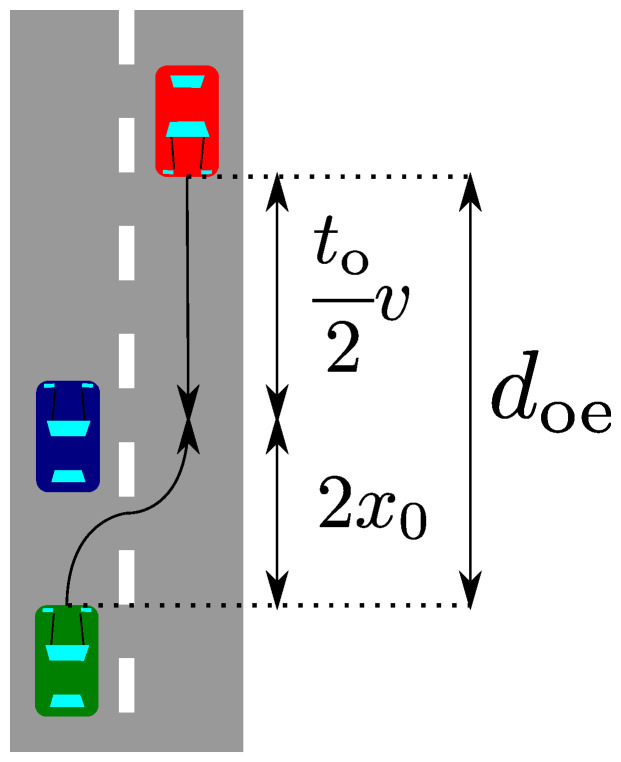
The distance that the ego and oncoming vehicle have moved by the collision of both vehicles.

**Figure 7 sensors-21-02659-f007:**
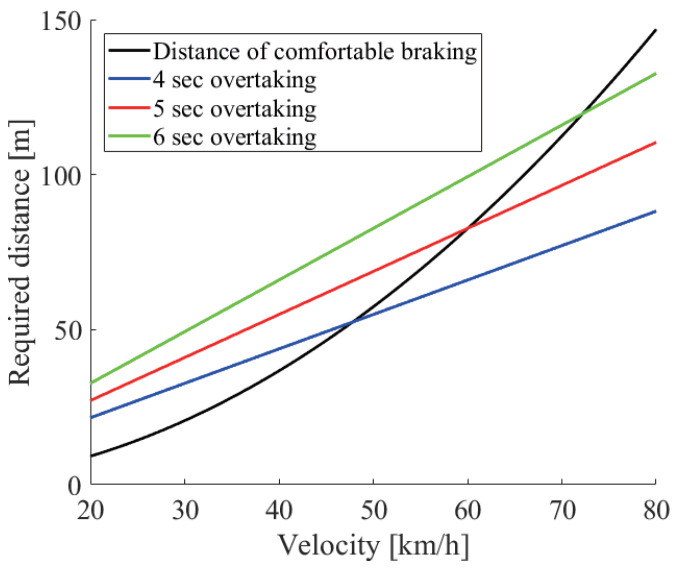
The distance required for the driving path and comfortable braking.

**Figure 8 sensors-21-02659-f008:**
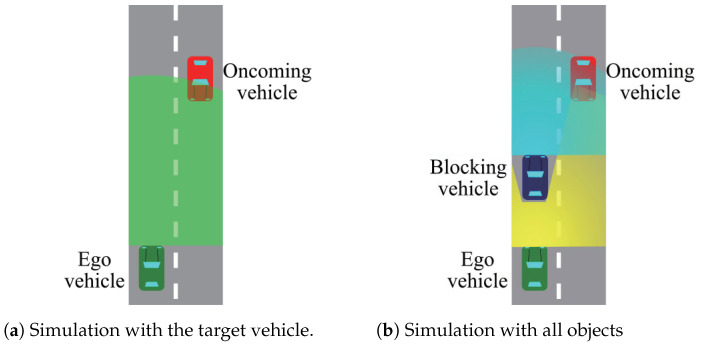
The two configurations for the recognition score.

**Figure 9 sensors-21-02659-f009:**
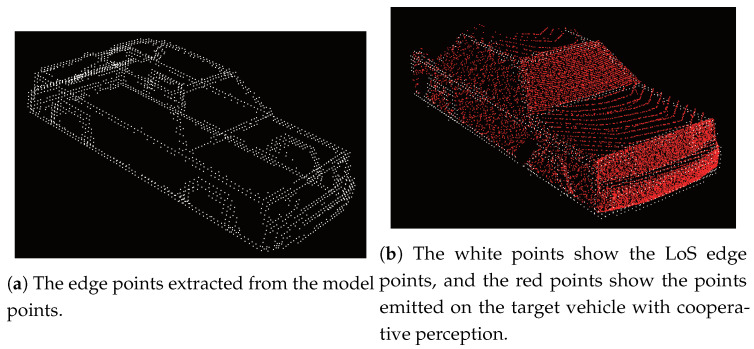
Examples of the point cloud used in the recognition process.

**Figure 10 sensors-21-02659-f010:**
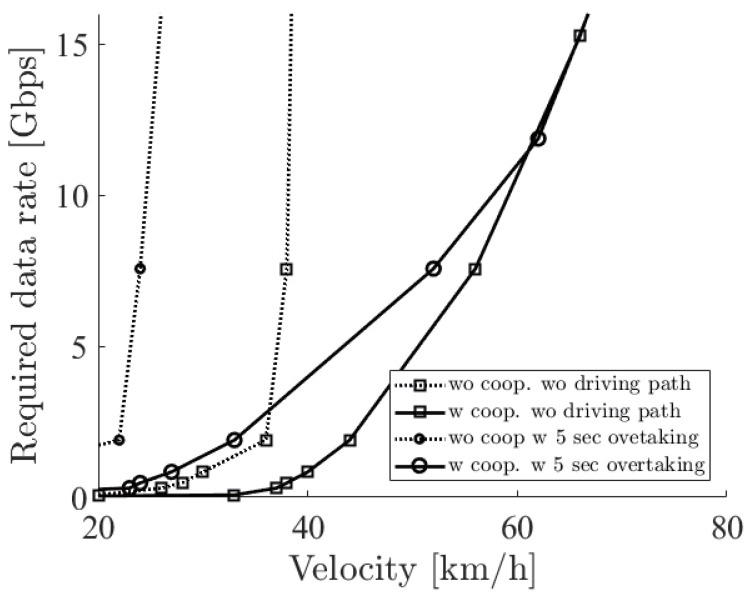
The required sensor data rate with the options of cooperative perception and the driving path when the distance between the ego vehicle and the blocking vehicle is 5 m.

**Figure 11 sensors-21-02659-f011:**
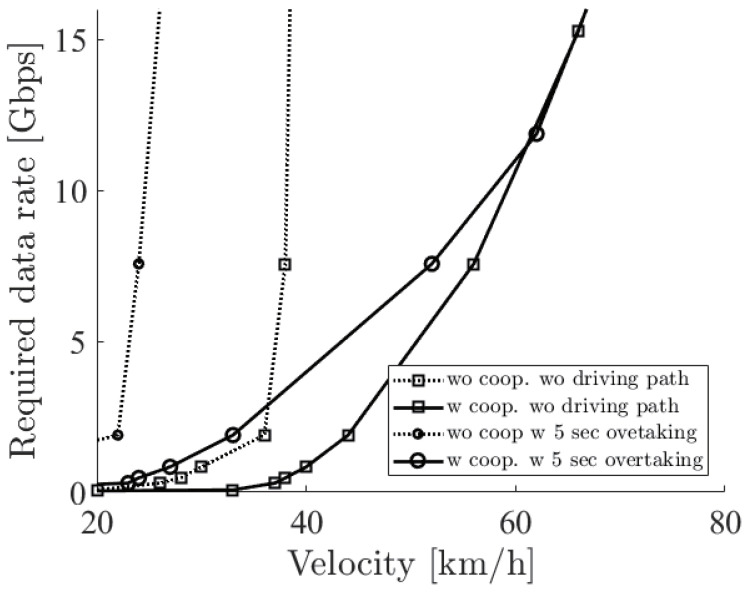
The required sensor data rate with the options of cooperative perception and 5 s overtaking when the distance between the ego vehicle and the blocking vehicle is 5, 10, and 15 m.

**Figure 12 sensors-21-02659-f012:**
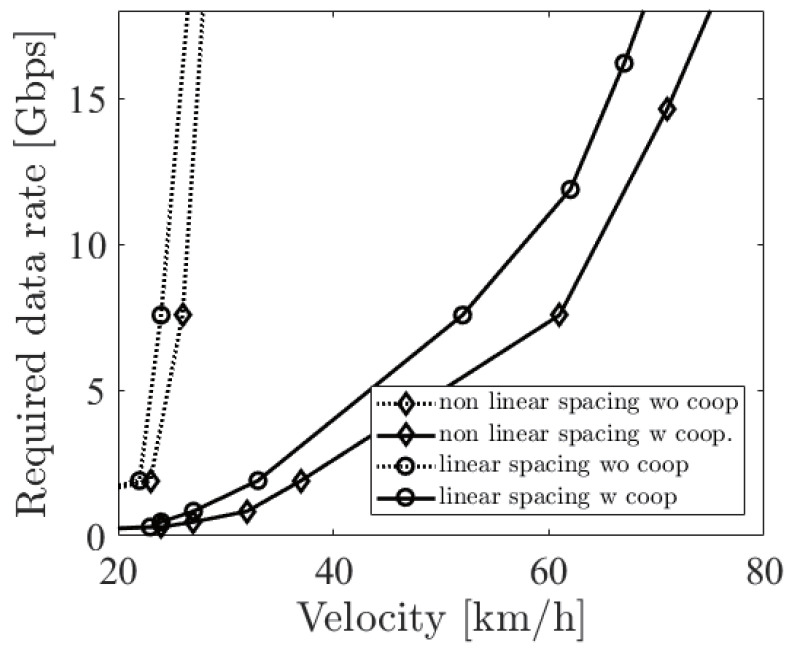
The required sensor data rate with the options of a linear and a non-linear spacing LiDAR sensor at $d_{\mathrm{be}}=10$ m.

**Figure 13 sensors-21-02659-f013:**
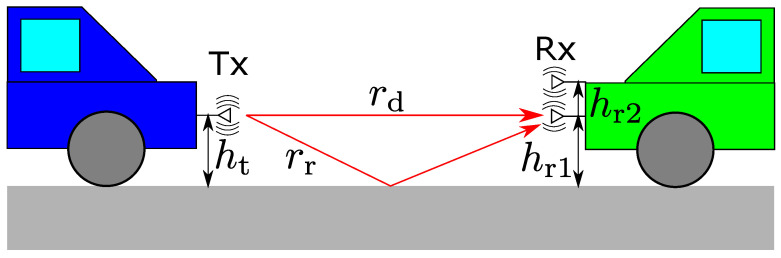
V2V two-ray ground reflection channel model.

**Figure 14 sensors-21-02659-f014:**
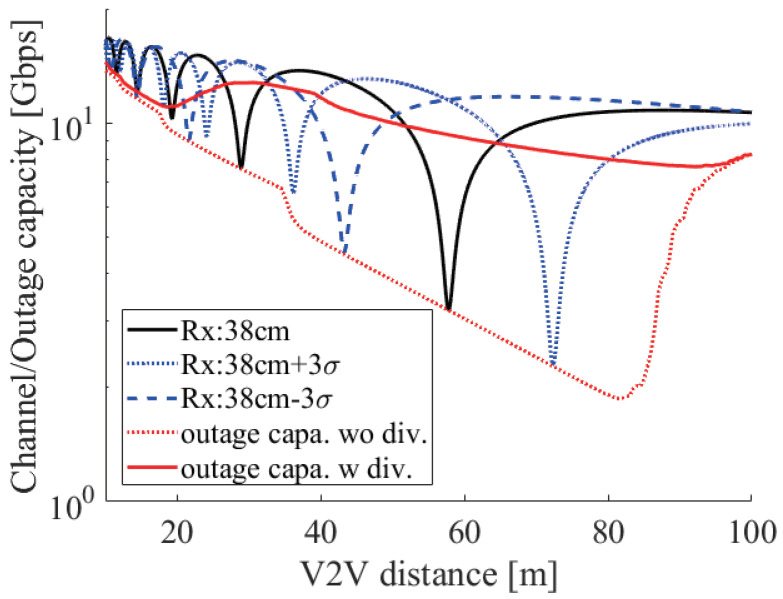
The channel capacity under the two-ray ground reflection model with and without the antenna vibration and the 0.01% outage capacity with and without height diversity at 60 GHz.

**Figure 15 sensors-21-02659-f015:**
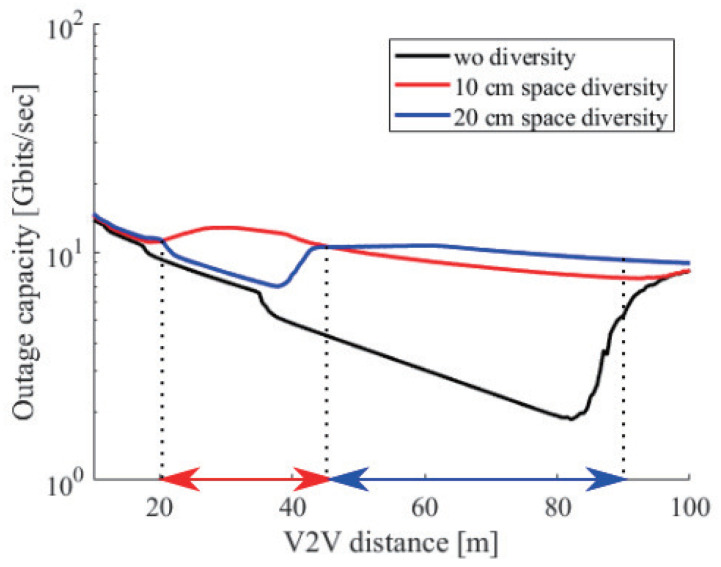
The 0.01% outage capacity with height diversity under $h_{t1},\;h_{r1}=0.38\;m$, $h_{r2}=0.48\;m$, and $h_{r2}=0.58\;m$ and without diversity at 60 GHz.

**Figure 16 sensors-21-02659-f016:**
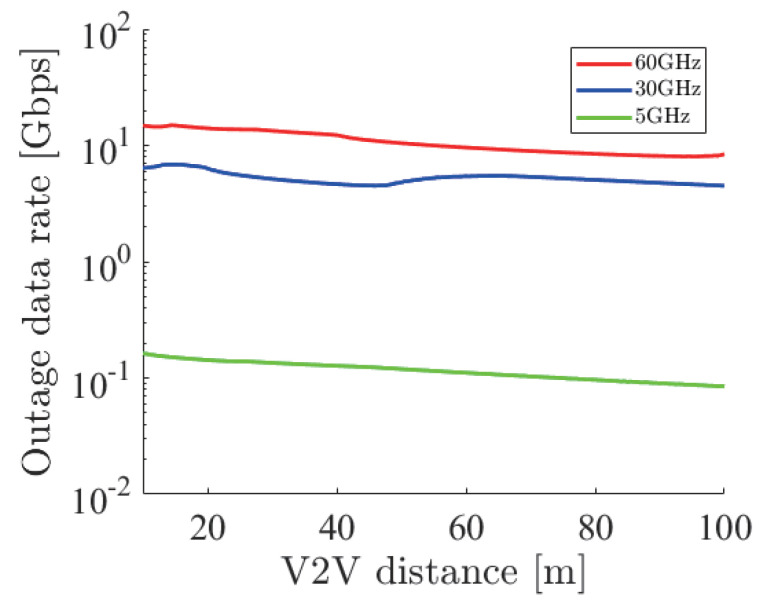
The 0.01% outage capacity with height diversity for 5, 30, and 60 GHz.

**Figure 17 sensors-21-02659-f017:**
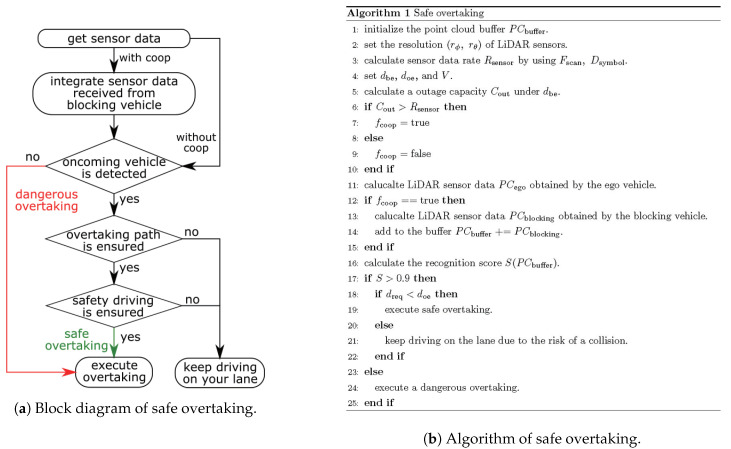
Description of the whole process in the simulation.

**Figure 18 sensors-21-02659-f018:**
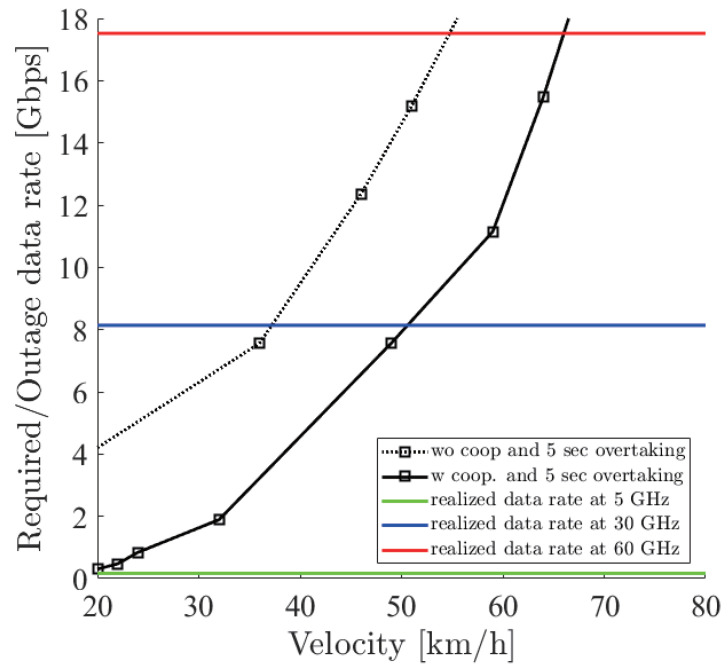
Required data rate and 0.01% outage data rate realized by the 5, 30 and 60 GHz bands under $d_{\mathrm{be}}=5$ m.

**Figure 19 sensors-21-02659-f019:**
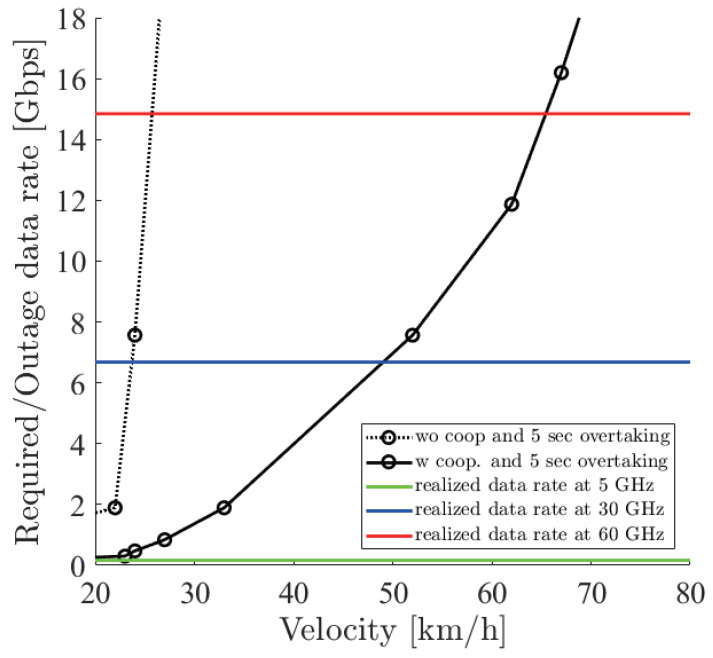
Required data rate and 0.01% outage data rate realized by the 5, 30 and 60 GHz bands under $d_{\mathrm{be}}=10$ m.

**Figure 20 sensors-21-02659-f020:**
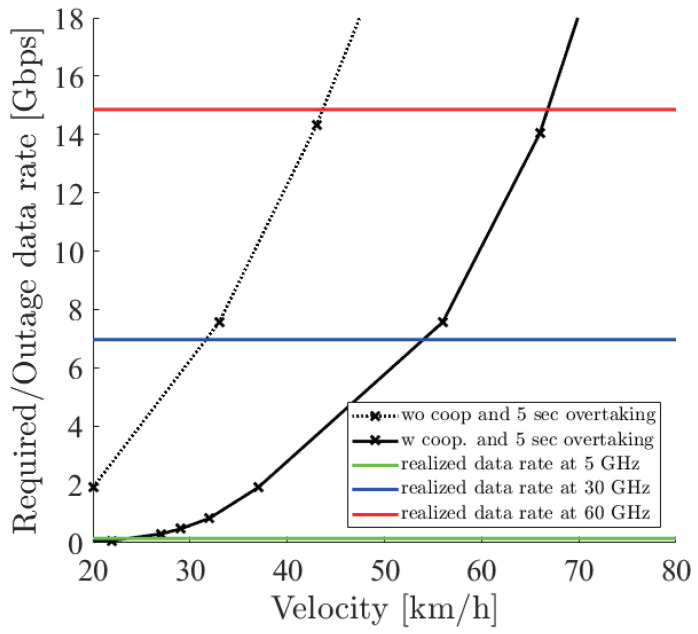
Required data rate and 0.01% outage data rate realized by the 5, 30 and 60 GHz bands under $d_{\mathrm{be}}=15$ m.

**Figure 21 sensors-21-02659-f021:**
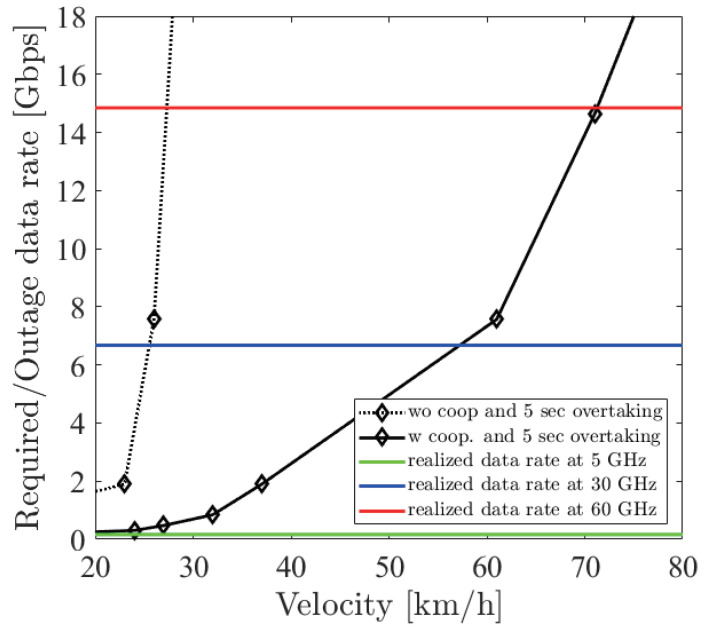
Required data rate using a non-linear spacing LiDAR sensor and the 0.01% outage data rate realized by the 5, 30, and 60 GHz bands under $d_{\mathrm{be}}=10$ m.

**Table 1 sensors-21-02659-t001:** Simulation parameters.

**LiDAR Parameters**	
**Parameter**	**Value**
Location	Vehicle’s roof +20 cm
Range	200 m
Elevation Angle Range	±15°
Elevation Angle Resolution $\left(r_{\phi }\right)$	[0.2°,0.1°,0.08°, 0.06°,0.04°,0.02°]
Azimuth Angle Range	180°
Azimuth Angle Resolution $\left(r_{\theta }\right)$	[0.2°,0.1°,0.08°, 0.06°,0.04°,0.02°]
Non-Linear Spacing	dense spacing (−7.5°–7.5°) sparse spacing (otherwise)
Return Mode	Strongest
Scan Period	20 Hz
Data Size of One Point	16 bit (coordinate)
	+ 12 bit (power)
**V2V System Parameters in the [5, 30, 60] GHz Bands**	
**Parameter**	**Value**
Height of Tx($h_{t}$)	38 cm
Height of Rx1($h_{r1}$)	38 cm
Height of Rx2($h_{r2}$)	[98, 48, 43] cm
Height of Rx3($h_{r3}$)	[182, 62, 50] cm
Transmitted Power	10 dBm
Boresight Gain	[4.3, 20, 26] dB
Antenna Aperture Size	2.6 cm × 2.6 cm
Polarization	vertical
Vertical Antenna Vibration Model	Gaussian(σ=3.2 cm)
Bandwidth	[10, 500, 1000] MHz
Antenna Diversity	selection diversity
Noise Figure	10 dB

## Data Availability

Not applicable.

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
