# Peer review of "Automated Driving with Cooperative Perception Using Millimeter-Wave V2V Communications for Safe Overtaking"

_sensors, 2021, doi:10.3390/s21082659_

Round 1
Reviewer 1 Report
The paper analyzed the required sensor data rate to be exchanged for the cooperative perception to enable a new level of safe and reliable automated driving in overtaking scenarios. The author calculated the required sensor data rate using LiDAR recognition and vehicle movement to adopt realistic assumptions for safe overtaking in automated driving. Furthermore, the authors also analyzed sensor data rate and outage data rate for conventional V2V communication and millimeter-wave communication. There are some inconsistencies and unclear descriptions. The current version of the manuscript needs to improve substantially to justify the proposed approach.
Comments # 1:
In the introduction section, the paper needs improvement in highlighting the problem statement. The paper lacks highlighting the main problem or research gap that the work is trying to address in the current version. The authors should explain the missing gap concerning the existing literature and why it is essential to address those gaps, which ultimately leads to the paper's objective that is trying to be achieved. The contributions of the paper should be listed in short, concise points.
Comments #2:
The scope of the literature survey of the paper is too small and can be improved. I suggest the authors rearrange the introduction and have a separate section for "Related work.". The contributions are not well defined. Also, the gap between the proposed work and the existing works is not well highlighted. The limitations of the current works need to be highlighted.
Comments #3:
From lines 70~ 90 author describes the paper's organization; however, the various chapters' topics do not match the description.
Concern # 4:
The organization of the paper is inadequate. The author should consider organizing the paper in the following sections: Introduction, Related works, System model, Proposed algorithm, Simulation/Results, and Conclusion.
Concern # 5:
Figure 8, Figure 13, and Figure 14 should be under the simulation/results section.
Concern # 6:
What is the main algorithm or framework proposed in this paper? Can the author present a pseudo code for the proposed framework or algorithm?
Concern # 7:
The author states that a ray-tracing 265 simulation of LiDAR sensor data is adopted. Can the author compare his results with different LiDAR sensor data and show the performance difference.
Concern # 8:
The explanation for height diversity is not clear. Maybe the author should provide a reference to support his claim.
Concern # 9:
In the simulation, the author only compares the required sensor data rate with the outage data rate, which is just a different configuration of the proposed algorithm. The author should compare the proposed scheme with some current state of the art methods.

Reviewer 2 Report
Is it "automated" or "autonomous" vehicle? Since both terms are found in.
All figures must include labels and units as well as clear caption. Please apply (a), (b), etc. for more than two pictures under one caption.
Does figure 1 corresponds to safe autonomous driving? What is the distance between blue and green vehicles? Does the blue car make a shadow for overtaking taking into account that distance is more than several meters.
Please remove figures from the Conclusions.
From my point of view, conclusions do not bring clearness. In order to make clear please add supporting information about speed of oncoming vehicle as well as blocking vehicle.
Is blocking vehicle is stopped vehicle? If yes, what about acceleration? Have you evaluated it? If blocking vehicle decreases speed, how it is estimated and how distance changes of green vehicle?
What impact and effect of sidewalls of tall buildings in narrow downtown streets?
No comparison of the results to the state-of-art.
"Related works" section is missing.
References are too old.
No experiments even at laboratory level.
Round 2
Reviewer 2 Report
Almost all comments were addressed.
Author Response
Thanks to your comments, my paper gets better.
If it is all, I would like to end.